# Interaction Between Lactic Acid Bacteria and Acetic Acid Bacteria in Sichuan Bran Vinegar: Impact on Their Growth and Metabolites

**DOI:** 10.3390/foods14091471

**Published:** 2025-04-23

**Authors:** Jianlong Li, Jie Wu, Meiling Tu, Xue Xiao, Kaidi Hu, Qin Li, Ning Zhao, Aiping Liu, Xiaolin Ao, Xinjie Hu, Shuliang Liu

**Affiliations:** 1College of Food Science, Sichuan Agricultural University, Ya’an 625014, China; jlli999@foxmail.com (J.L.); 13111899179@163.com (J.W.); tmling1213@163.com (M.T.); kaidi666@outlook.com (K.H.); 17888820100@163.com (Q.L.); zhaoning@sicau.edu.cn (N.Z.); huavslin@163.com (X.A.); xinjiehu@sicau.edu.cn (X.H.); 2Key Laboratory of Agricultural Product Processing, Nutrition Health (Co-Construction by Ministry and Province), Ministry of Agriculture and Rural Affairs, Ya’an 625014, China; xiaoxue0420@163.com

**Keywords:** Sichuan bran vinegar, *Acetobacter pasteurianus*, *Lactobacillus amylovorus*, interaction, volatile compounds

## Abstract

Microbial interactions are essential for maintaining the stability and functionality of microbiota in fermented foods. In this study, representative strains of predominant lactic acid bacteria and acetic acid bacteria in Sichuan bran vinegar were selected, and their interactions in a simulated solid-state fermentation system were investigated. The results reveal that the biomass of *A. pasteurianus* LA10 significantly increased in both the co-culture and the pure culture, whereas the biomass of *L. amylovorus* LL34 in the co-culture (6.44 ± 0.30 lg CFU/g) was significantly lower than that in the pure culture (7.28 ± 0.30 lg CFU/g) (*p* < 0.05), indicating a partially harmful symbiosis between these two strains. The metabolic analysis shows that total acid (21.82 mg/g) and acetic acid (9.53 mg/g) contents in the co-culture were lower than those in the pure culture of LA10, suggesting that LL34 inhibited the acid-producing activity of LA10 to some extent. The interaction between the two bacteria also influenced the production of volatile compounds and non-volatile compounds, as revealed by GC-MS and untargeted UHPLC-MS/MS, respectively. Significant enrichment of acid and amino acid metabolism pathways was observed in the co-culture, revealing the impact of bacterial interactions on flavor development. This study provides valuable insights into the advancement of vinegar brewing technology.

## 1. Introduction

Cereal vinegar is a vital acidic condiment in the daily Chinese diet, with a history spanning over 3000 years [1,2]. Various types of vinegar have been developed, particularly Sichuan Baoning vinegar, Zhenjiang aromatic vinegar, Shanxi aged vinegar, and Fujian Monascus vinegar [3]. Traditional production involves solid-state fermentation; however, a more efficient approach involving liquid-state fermentation followed by solid-state fermentation is now preferred [4]. Sichuan Baoning vinegar is a representative Sichuan bran vinegar that distinguishes itself from other vinegar due to its whole process of traditional solid-state fermentation using uncooked wheat bran as the primary raw material. This vinegar undergoes saccharification, alcohol fermentation, and acetic acid fermentation simultaneously in a single fermentation pond, with Daqu (also called great koji, a traditional fermentation starter made from a blend of grains that have been moistened and allowed to ferment for several days) incorporated with Chinese herbs serving as a saccharifying agent [5].

Lactic acid and acetic acid are the predominant organic acids in solid-state vinegar fermentation, comprising over 90% of the total organic acid content [6,7,8]. Lactic acid is characterized by its mildness, while acetic acid imparts a more pungent flavor. The acid ratio plays a crucial role in determining the sensory characteristics of vinegar [9]. Sichuan bran vinegar is notable for its higher lactic acid content than acetic acid [10,11]. As is known, the microbial composition of vinegar is complex, with microbial interactions essential for its stability and functionality. The dominant bacteria in the solid-state fermentation of vinegar differ at different periods, such as mold in the early stage of fermentation (starch saccharification fermentation), yeast in the middle stage (alcohol fermentation), and acetic acid bacteria in the late stage (acetic acid fermentation). At the same time, there are other dominant bacteria, such as *Lactobacillus*, *Bacillus*, *Stenotrophomonas*, *Methyloversatilis*, and *Amycolatopsis*, etc., among which lactic acid can produce lactic acid to neutralize the irritation of acetic acid [12]. *Bacillus* can secrete a variety of hydrolases that play an important role in liquefaction and saccharification and can produce organic acids through the tricarboxylic acid cycle [13]. However, a comprehensive understanding of these interactions in vinegar remains limited [14]. Lactic acid and acetic acid are key metabolites produced by lactic acid bacteria and acetic acid bacteria, respectively. Metagenomic sequencing revealed *Lactobacillus amylovorus* (*L. amylovorus*) and *Acetobacter pasteurianus* (*A. pasteurianus*) as dominant strains in *Cupei* (the mixture of ingredients used for cereal vinegar fermentation) during Sichuan Baoning vinegar fermentation [15].

In this study, *L. amylovorus* LL34 and *A. pasteurianus* LA10 [16], isolated from Sichuan Baoning vinegar for their high lactic acid and acetic acid yields and robust growth capacities, were employed as representative strains. This study aimed to investigate the interaction between these two strains during fermentation using a simulated fermentation system and its impact on strain growth, acid production, and volatile and non-volatile compounds. This study offers insights into microbial interactions during fermentation and provides theoretical guidance for enhancing the quality of Sichuan bran vinegar.

## 2. Materials and Methods

### 2.1. Strains and Growth Conditions

*L. amylovorus* LL34 and *A. pasteurianus* LA10, isolated from *Cupei* of Sichuan Baoning vinegar, were maintained at the Laboratory of Microbiology, College of Food Science, Sichuan Agricultural University. The strains were individually cultured in 5 mL of MRS broth (10 g peptone, 8 g beef extract, 4 g yeast extract, 20 g glucose, 2 g K_2_HPO_4_, 5 g CH_3_COONa, 2 g ammonium citrate, 0.2 g MgSO_4_·7H_2_O, 0.05 g MnSO_4_·H_2_O, 1 mL Tween 80, pH 6.2–6.5, and 1 L H_2_O; *L. amylovorus* LL34) and 5 mL of GYE broth (1% glucose, 1% yeast extract, and 3% ethanol; *A. pasteurianus* LA10). *L. amylovorus* LL34 was statically incubated at 37 °C for 24 h, while *A. pasteurianus* LA10 was cultured in a shaking incubator at 30 °C for 72 h at 160 rpm.

### 2.2. Interaction Between Lactic Acid Bacteria and Acetic Acid Bacteria

#### 2.2.1. Solid-State Fermentation and Sampling

A *Cupei* stimulation medium comprising wheat bran (30%), wheat flour (20%), and MRS broth (50%) was developed for solid-state fermentation. The medium was sterilized at 121 °C and subsequently supplemented with 2% ethanol, 1.3% lactic acid, and 0.3% acetic acid. The concentrations of ethanol and organic acids were based on our previous report [17]. *L. amylovorus* LL34 (10^6^ CFU/g, obtained by MRS broth culture) and *A. pasteurianus* LA10 (4 × 10^5^ CFU/g, obtained by GYE broth culture) were co-inoculated into 200 g of the medium and statically incubated at 37 °C for 60 h. Samples were collected at 12 h intervals (0, 12, 24, 36, 48, and 60 h) after thoroughly mixing the medium. Each strain was individually inoculated as a control, and triplicate parallel experiments were conducted for each experimental group.

#### 2.2.2. Enumeration of Strains

The number of *L. amylovorus* LL34 and *A. pasteurianus* LA10 in each sample was determined by plating on an MRS agar plate [18]. The two bacterial strains were distinguished based on colony morphology. The colony of *L. amylovorus* LL34 was larger and irregularly circular, while the colony of *A. pasteurianus* was smaller and exhibited a regular circular shape.

#### 2.2.3. Determination of Total Acid, Lactic Acid, and Acetic Acid Content

Five grams of *Cupei* was mixed with 45 mL of sterile water and kept at 60 °C for 1 h. After cooling to 30 °C, the mixture was centrifuged at 10,000 g for 10 min, and the supernatant was collected for analysis. The total acid content was tested by titration with 1.0 mol/L NaOH using phenolphthalein as an indicator and expressed as g/100 g lactic acid.

Organic acid contents were analyzed with modifications from a previous report [17]. Briefly, 4 mL of supernatant was mixed with 1 mL of potassium ferrocyanide (106 g/L) and 1 mL of zinc sulfate (300 g/L), followed by centrifugation at 10,000 g for 10 min. The supernatant was extracted using Sep-Pak C18 cartridges, filtered through a 0.22 μm membrane, and analyzed by HPLC. A Hichrom Alltech OA-1000 column (300 × 6.5 mm, 9 μm) was employed with 9 mmol/L H_2_SO_4_ as the mobile phase at a flow rate of 0.6 mL/min. The injection volume was 10 μL, and the UV detector was set at 210 nm. A calibration curve was established using organic acid standard solutions, with concentration calculated versus peak area using the least-squares method.

#### 2.2.4. Volatile Compounds

Volatile compounds were analyzed using gas chromatography–mass spectrometry (GC–MS; 5975C/6890N; Agilent, CA, USA) with an HP-5MS column (60 m × 0.32 mm, 1 μm) according to a previous report [17]. Compound identification was performed by comparing obtained data with the National Institute of Standards and Technology (NIST) Library 2020 (similarity index > 80%) and literature references. Methyl heptanoate (0.001 mg/mL) was used as an internal standard for semi-quantification.

#### 2.2.5. Non-Volatile Compounds

Non-volatile compounds were determined using untargeted metabolomics via the UHPLC–Exactive HF-X system by Majorbio Co., Ltd. (Shanghai, China), following the report by Kang et al. [19].

### 2.3. Statistical Analysis

All analyses were performed in triplicate, and experimental data are presented as the mean ± standard deviation. Statistical analysis was performed via SPSS 22.0 (SPSS, Chicago, IL, USA). One-way analysis of variance (ANOVA) was performed with a 95% confidence interval (*p* < 0.05), and the mean comparison was performed via the least significant difference (LSD) test.

## 3. Results and Discussion

### 3.1. Enumeration of Strains in Pure Culture and Co-Culture

The growth of *L. amylovorus* LL34 and *A. pasteurianus* LA10 was measured in a pure culture and a co-culture (Figure 1A). The growth trends of *A. pasteurianus* LA10 in both cultures were similar, characterized by rapid initial growth within the first 24 h, followed by slower growth. The final biomass was 8.37 lg CFU/g in the pure culture and 8.50 lg CFU/g in the co-culture. In contrast, *L. amylovorus* LL34 demonstrated distinct growth behaviors: in the pure culture, its biomass increased significantly (*p* < 0.05), whereas, in the co-culture, the increase was not significant (*p* > 0.05), with the biomass increasing from 6.07 to only 6.44 lg CFU/g. These findings indicate that the presence of *L. amylovorus* LL34 had minimal impact on the growth of *A. pasteurianus* LA10, whereas the presence of *A. pasteurianus* LA10 may inhibit the growth of *L. amylovorus* LL34.

### 3.2. Changes in Total Acid Content in Pure Culture and Co-Culture

The total acid content in both the pure culture and the co-culture was measured (Figure 1B). In the pure culture of *A. pasteurianus* LA10 and the co-culture, the total acid content tended to increase as fermentation progressed. Specifically, the pure culture of *A. pasteurianus* LA10 increased from 13.15 to 26.16 mg/g, while the co-culture increased from 15.39 to 21.82 mg/g. Conversely, the pure culture of *L. amylovorus* LL34 exhibited no significant change in the total acid content, remaining stable from 14.94 mg/g to 14.79 mg/g (*p* > 0.05). These findings suggest that the acidogenic metabolism of *A. pasteurianus* LA10 was partially inhibited in the co-culture.

Different capital letters denote significant differences in the same strain at varying fermentation times, while different lowercase letters indicate significant differences between the different cultures at a single time point (*p* < 0.05). Abbreviations: ML, the pure culture of *L. amylovorus* LL34; MA, the pure culture of *A. pasteurianus* LA10; CL, *L. amylovorus* LL34 in the co-culture; CA, *A. pasteurianus* LA10 in the co-culture; L, the pure culture of *L. amylovorus* LL34; A, the pure culture of *A. pasteurianus* LA10; and CO, co-culture.

### 3.3. Changes in Lactic Acid and Acetic Acid Contents in Pure Culture and Co-Culture

The lactic acid and acetic acid contents in the pure culture and the co-culture were measured (Figure 1C,D). The results indicate that the lactic acid and acetic acid contents in the pure culture of *L. amylovorus* LL34 remained stable throughout fermentation, consistent with the total acid content. Conversely, both the pure culture of *A. pasteurianus* LA10 and the co-culture exhibited significant changes in lactic acid and acetic acid contents (*p* < 0.05). Specifically, the lactic acid content in the pure culture of *A. pasteurianus* LA10 decreased from 11.14 to 8.47 mg/g and from 10.91 to 10.02 mg/g in the co-culture. The acetic acid content in the pure culture of *A. pasteurianus* LA10 increased from 3.92 to 12.20 mg/g and from 3.62 to 9.53 mg/g in the co-culture. The decline in lactic acid content during later fermentation stages in the pure culture of *A. pasteurianus* LA10 and the co-culture may be attributed to the utilization of lactic acid by *A. pasteurianus* [20]. Notably, the acetic acid level was found to be lower in the co-culture compared to the pure culture of *A. pasteurianus* LA10 after 12 h incubation, indicating potential metabolic regulation of *A. pasteurianus* LA10 by *L. amylovorus* LL34.

### 3.4. Volatile Compounds in Pure Culture and Co-Culture

A total of 67 volatile compounds were identified using GC-MS, including acids (6), alcohols (8), aldehydes (11), esters (11), ketones (13), heterocycles (3), terpenes (2), and others (13) (Table 1). *A. pasteurianus* LA10 in the pure culture exhibited the most detection (39), followed by the co-culture (36), and *L. amylovorus* LL34 in the pure culture (34). To investigate the differences in volatile compounds in different cultures, an orthogonal partial least-squares discriminant analysis (OPLS-DA) model was established (Figure 2A), with 200 permutation tests conducted (Figure 2B). The model demonstrated a goodness of prediction (Q^2^) exceeding 0.5. The score plot revealed significant differences among the cultures.

Acids are important metabolites during fermentation, constituting 14.99–36.30% of volatile compounds. Acetic acid was the most abundant, and its relative content in the pure culture of *A. pasteurianus* LA10 (810.72 ± 235.26 μg/kg) exceeded that in the co-culture (607.97 ± 205.27 μg/kg), aligning with acetic acid determination.

The relative content of alcohols ranged from 8.54% to 26.49%, with ethanol being the predominant alcohol. The relative content of ethanol was the highest in the pure culture of *L. amylovorus* LL34 (506.91 ± 56.92 μg/kg), followed by the co-culture (393.39 ± 90.212 μg/kg) and the pure culture of *A. pasteurianus* LA10 (193.91 ± 30.37 μg)/kg. The findings indicate that the ethanol utilization ability of *A. pasteurianus* LA10 in the co-culture was influenced by *L. amylovorus* LL34, highlighting the impact of the co-culture on the growth and metabolism of *A. pasteurianus* LA10.

The relative contents of aldehydes, esters, and ketones ranged from 0.59% to 12.95%, 0.23% to 12.7%, and 2.53% to 8.28%, respectively. The relative content of benzaldehyde in the co-culture was the lowest. Additionally, flavor substances such as ethyl acetate and 3-hydroxy-2-butanone [21] were found to be lower in the co-culture compared to the pure culture of *A. pasteurianus* LA10. 3-Hydroxy-2-butanone, also known as acetoin, served as a precursor in the synthesis of tetramethylpyrazine in vinegar [5].

Heterocyclic compounds and terpenoids were detected. Heterocyclic compounds showed a higher relative abundance in all three cultures, while terpenoids had relatively lower proportions. Notably, the co-culture exhibited a higher relative content of 2-pentylfuran compared to the other two pure cultures. This compound is commonly found in fermented foods, imparting a fruity and milky aroma [22,23].

### 3.5. Non-Volatile Compounds in Pure Culture and Co-Culture

#### 3.5.1. Metabolite Profile

A total of 13,343 peaks (7055 in positive mode and 6288 in negative mode) was observed using untargeted metabolomics analysis. Through annotation via the Human Metabolome Database (HMDB), 2465 metabolites were identified, predominantly comprising lipids and lipid molecules (631), organic acids and their derivatives (545), organic heterocyclic compounds (371), organooxygen compounds (305), and benzenes (226) (Figure 2C).

#### 3.5.2. Differential Metabolite Analysis

The metabolite profiles of the pure culture and the co-culture were analyzed using principal component analysis (PCA) (Figure 3). Quality control (QC) sample clustering demonstrates the stability and reproducibility of the detection procedure. In positive mode, the first and second principal components contributed 15.00% and 13.80% of the variance, respectively. Similarly, in negative mode, the first and second principal components contributed 15.50% and 14.40% of the variance, respectively. The metabolite compositions in the co-culture and the pure culture of *A. pasteurianus* LA10 were similar, as evidenced by their overlapping positions in the PCA plot.

OPLS-DA was employed to obtain the variable influence on projections (VIPs) and *p*-values of the Student’s *t*-tests. Differential metabolites were identified as those with a VIP value > 1 and *p* < 0.05. Comparative metabolite analysis reveals 330 differential metabolites between the co-culture and the pure culture of Acetobacter LA10 and 441 differential metabolites between the co-culture and the pure culture of *L. amylovorus* LL34 (Figure 4).

Differential metabolites with top VIP values were selected between the co-culture and the pure culture of *A. pasteurianus* LA10 (Table 2). The upregulated metabolites after co-culture included phenyllactic acid, 3-phenyllactic acid, 4-vinylphenol, 6-hydroxyhexanoic acid, etc. Phenyllactic acid, produced by most lactic acid bacteria, exhibits broad-spectrum antibacterial properties [24] and serves as a metabolic marker for vinegar identification and classification [25]. Similar to phenyllactic acid, 6-hydroxyhexanoic acid demonstrates antioxidant and anti-inflammatory properties [26]. 4-Vinylphenol, a characteristic flavor compound, significantly contributes to the flavor formation of other fermented foods [27].

Differential metabolites with top VIP values were selected between the co-culture and the pure culture of *L. amylovorus* LL34 (Table 3). The upregulated metabolites after co-culture included scopoletin, isofraxidin, 5-Hydroxymethyl-2-furancarboxaldehyde, etc. Scopoletin and isofraxidin are both coumarin compounds with favorable biological characteristics [28,29] and can be isolated from plants. 5-Hydroxymethyl-2-furancarboxaldehyde is a major product of the Maillard reaction and can serve as an indicator of the adulteration of food products with acid-converted inverted syrups [30].

#### 3.5.3. KEGG Metabolic Pathway Analysis

The metabolites were searched for in the KEGG database and enriched in the KEGG pathways. A total of 1124 metabolic pathways were annotated in the metabolome, with amino acid metabolism comprising the highest proportion (186 pathways), followed by lipid metabolism (154 pathways) and the biosynthesis of other secondary metabolites (134 pathways).

KEGG enrichment analysis reveals that differential metabolites between the co-culture and the pure culture of *A. pasteurianus* LA10 were mainly associated with arachidonic acid metabolism, glycerophospholipid metabolism, and phenylpropanoid biosynthesis (Figure 5A), while differential metabolites between the co-culture and the pure culture of *L. amylovorus* LL34 were mainly associated with the biosynthesis of phenylpropanoids, D-amino acid metabolism, arginine, and proline metabolism, as well as nucleotide metabolism (Figure 5B).

The presence of acids, particularly organic acids and amino acids, in vinegar is crucial to its flavor. The significant enrichment of acids and amino acid metabolism pathways in the co-culture indicates its vital impact on flavor development.

## 4. Conclusions

In this study, a bottom-up approach was employed to examine the interactions between lactic acid bacteria and acetic acid bacteria in a simulated *Cupei* medium for Sichuan bran vinegar. The results reveal that the co-culture inhibited the growth and metabolism of *L. amylovorus* LL34 while minimally affecting the growth of *A. pasteurianus* LA10 but suppressing its metabolic processes. Microbial interactions influence both volatile and non-volatile metabolite profiles. This investigation provides foundational insights into how dominant microorganism interactions modulate the quality of Sichuan bran vinegar. Future research should investigate strain interactions at the transcriptomic and proteomic levels and explore potential interactions involving three or more strains.

## Figures and Tables

**Figure 1 foods-14-01471-f001:**
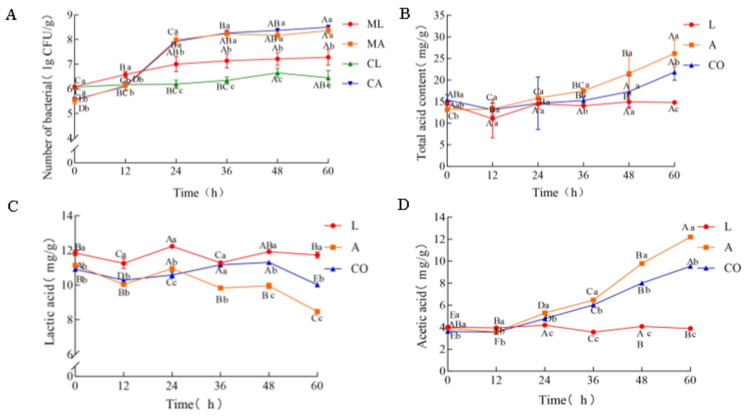
Index changes in pure culture and co-culture. (**A**) The number of *L. amylovorus* LL34 and *A. pasteurianus* LA10; (**B**) total acid content; (**C**) lactic acid content; (**D**) acetic acid content.

**Figure 2 foods-14-01471-f002:**
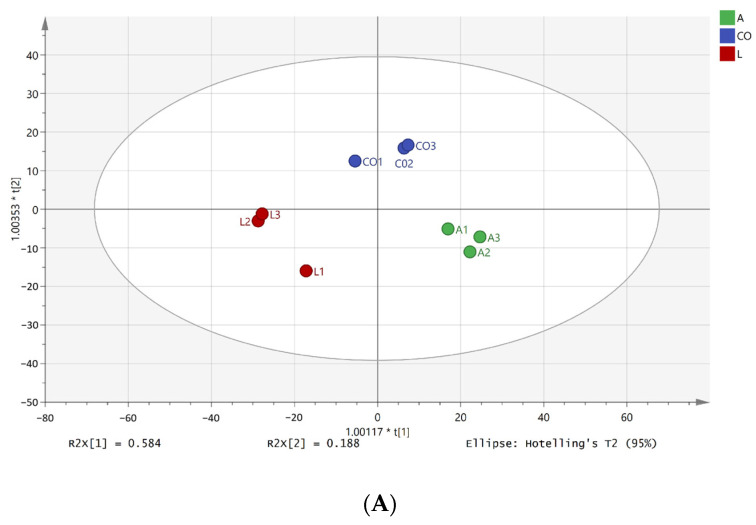
OPLS-DA (**A**), permutation test of OPLS-DA (**B**), and classification of compounds annotated in HMDB (**C**) in pure culture and co-culture. Note: (**A**,**B**): Difference analysis of volatile flavor components; (**C**): Metabonomic data analysis.

**Figure 3 foods-14-01471-f003:**
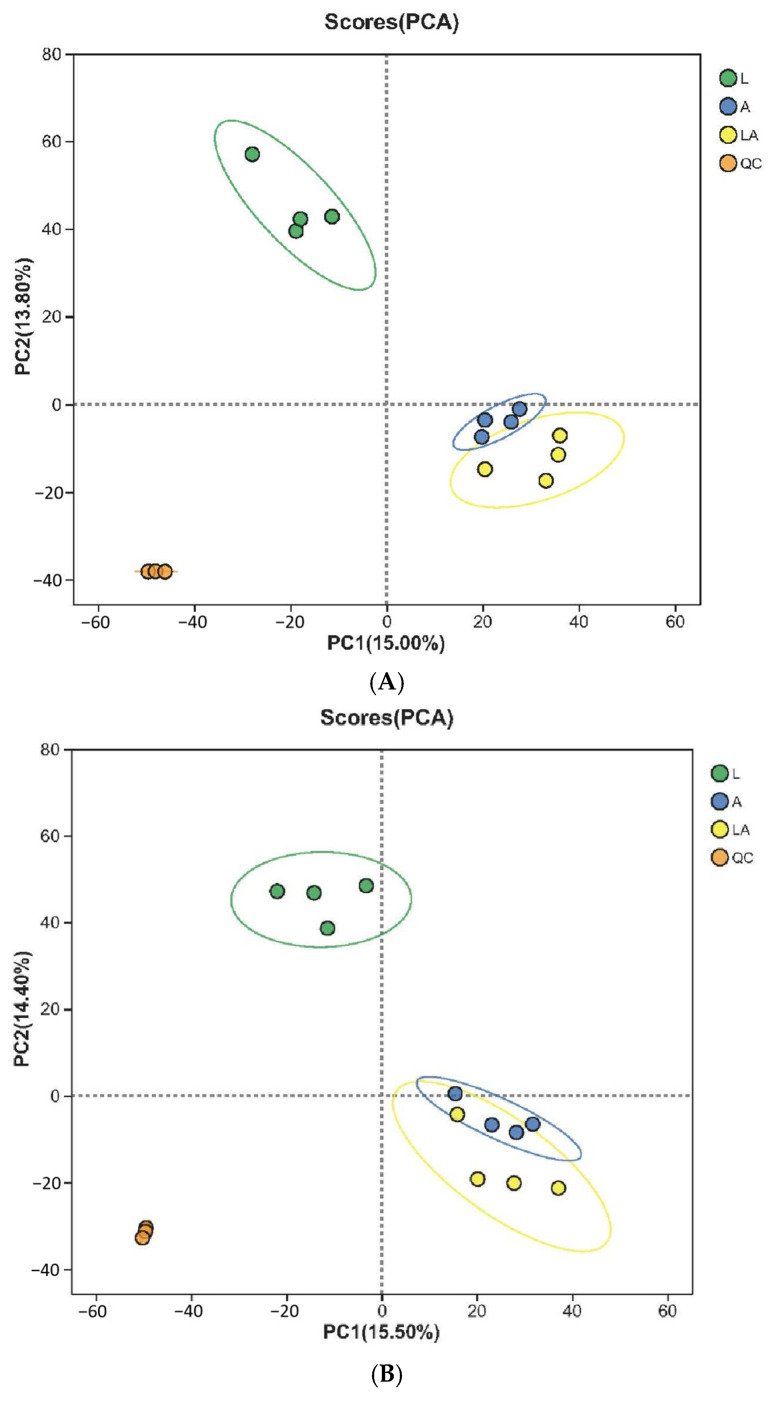
PCA plots of the pure culture and co-culture. (**A**) Positive mode; (**B**) negative mode. Abbreviation: L, pure culture of *L. amylovorus* LL34; A, pure culture of *A. pasteurianus* LA10; CO, co-culture.

**Figure 4 foods-14-01471-f004:**
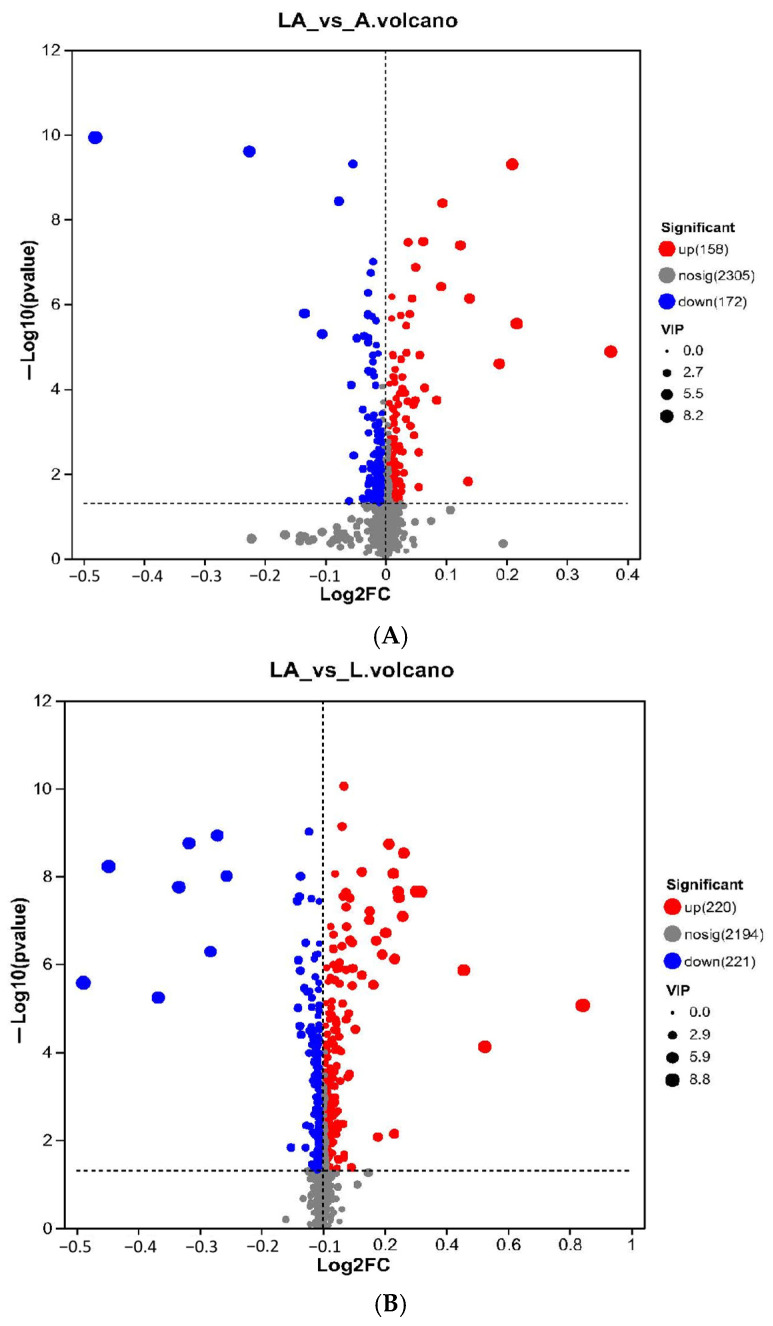
Volcano plot of differential metabolites. (**A**) LA10 co-culture and pure culture of *A. pasteurianus*; (**B**) LL34 co-culture and pure culture of *L. amylovorus*.

**Figure 5 foods-14-01471-f005:**
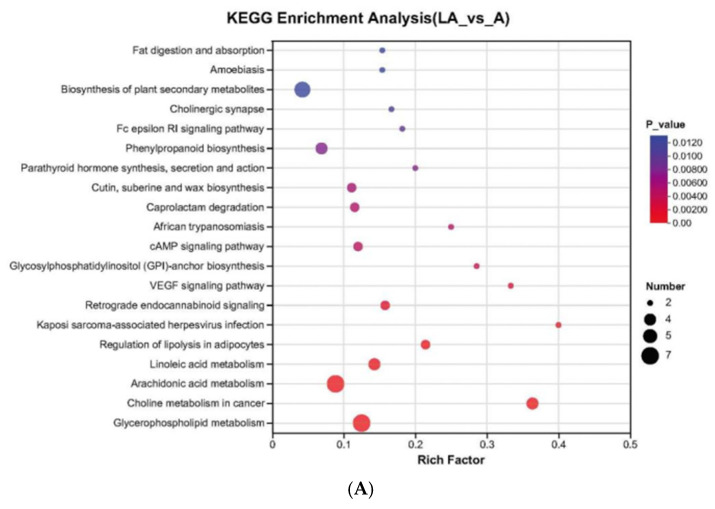
KEGG pathway enrichment analysis. (**A**) Differential metabolites between co-culture and pure culture of *A. pasteurianus* LA10; (**B**) differential metabolites between co-culture and pure culture of *L. amylovorus* LL34. Each bubble represents a metabolic pathway. The bubble size correlates with the abundance of metabolites enriched in the pathway.

**Table 1 foods-14-01471-t001:** Volatile compounds in pure culture and co-culture.

Class	No.	Name	Relative Contents (μg/kg)		
			Pure Culture of *L. amylovorus* LL34	Pure Culture of *A. pasteurianus* LA10	Co-Culture of *L. amylovorus* LL34 and *A. pasteurianus* LA10
Acids	1	Acetic acid	262.26 ± 66.25 ^b^	810.72 ± 235.26 ^a^	607.97 ± 205.27 ^ab^
	2	Hexanoic acid	/	62.84 ± 5.85 ^a^	58.88 ± 12.67 ^a^
	3	Pentanoic acid	31.63 ± 0.02 ^a^	/	/
	4	2-Methylbutanoic acid	/	3.72 ± 2.89 ^a^	/
	5	3-Methylbutanoic acid	/	3.31 ± 1.04 ^a^	/
	6	2-Methylpropanoic acid	/	2.28 ± 0.49 ^a^	/
Alcohols	7	Ethanol	506.91 ± 56.92 ^a^	193.91 ± 30.37 ^b^	393.39 ± 90.21 ^a^
	8	1-Hexanol	10.56 ± 0.75 ^a^	7.27 ± 0.25 ^c^	9.39 ± 0.03 ^b^
	9	Cyclohexanemethanol, 4-methyl-, cis-	/	4.13 ± 0.39 ^a^	/
	10	α-Methyl-benzenemethanol	/	2.2 ± 0.16 ^a^	/
	11	Benzenemethanol, 4-methyl-.alpha.-(1-methyl-2-propenyl)-, (R,R)-	/	0.21 ± 0.02 ^a^	/
	12	1-Octen-3-ol	/	/	9.02 ± 0.78 ^a^
	13	1-Pentanol	0.99 ± 0.2 ^a^	/	1.05 ± 0.02 ^a^
	14	2-Methyl-1,8-octanediol	1.09 ± 0.33 ^a^	/	/
Aldehydes	15	Benzaldehyde	115.46 ± 18.99 ^a^	10.5 ± 0.65 ^b^	9 ± 0.84 ^b^
	16	3-Furaldehyde	61.55 ± 6.4 ^a^	/	/
	17	Methylal	/	/	39.88 ± 32.55 ^a^
	18	5-Ethylcyclopent-1-enecarboxaldehyde	37.74 ± 2.99 ^a^	/	/
	19	Benzeneacetaldehyde	15.59 ± 2.45 ^a^	/	/
	20	Nonanal	8 ± 1.12 ^a^	1.27 ± 0.1 ^b^	2.02 ± 0.16 ^b^
	21	Heptanal	7.68 ± 0.36 ^a^	/	/
	22	Hexanal	6.69 ± 5.41 ^a^	/	/
	23	2-Butenal	/	2.54 ± 1.52 ^a^	4.8 ± 3.54 ^a^
	24	Decanal	0.94 ± 0.18 ^a^	/	/
	25	Isophthalaldehyde	0.41 ± 0.02 ^a^	/	/
Esters	26	Ethyl acetate	/	167.98 ± 15.94 ^a^	158.19 ± 26.88 ^a^
	27	Methyl 2-hydroxypropanoate	/	66.86 ± 19.81 ^a^	55.08 ± 45.24 ^a^
	28	Propyl 2-hydroxypropanoate	/	/	52.79 ± 59.97 ^a^
	29	Ethyl 2-(methylamino)acetate	/	28.21 ± 10.76 ^a^	35.5 ± 3.45 ^a^
	30	Ethyl 2-hydroxypropanoate	/	10.58 ± 0.31 ^b^	33.94 ± 2.26 ^a^
	31	Allyl acetate	/	4.49 ± 1.28 ^a^	/
	32	Phenol, 2,6-bis(1,1-dimethylethyl)-4-methyl-, methylcarbamate	3.02 ± 0.66 ^a^	/	/
	33	Hexyl acetate	/	2.07 ± 0.07 ^a^	1.77 ± 0.05 ^b^
	34	*2-Ethylhexyl hexyl sulfite*	1.52 ± 0.24 ^a^	0.8 ± 0.78 ^a^	3.24 ± 2.31 ^a^
	35	Methyl acetate	/	1.03 ± 0.38 ^a^	1.11 ± 0.3 ^a^
	36	Sulfurous acid, isobutyl pentyl ester	/	0.45 ± 0.22 ^a^	/
Ketones	37	Acetyl methyl carbinol	/	145.4 ± 31.4 ^a^	107.75 ± 20.93 ^a^
	38	Gamma-nonanoic lactone	15.45 ± 2.34 ^a^	20.3 ± 0.4 ^a^	19.12 ± 3.95 ^a^
	39	2-Heptanone	1.84 ± 0.47 ^b^	24.53 ± 1.48 ^a^	/
	40	5-Methyl-2-hexanone	25.03 ± 1.57 ^a^	2.29 ± 0.21 ^b^	13.63 ± 15.89 ^ab^
	41	2-Nonanone	1.04 ± 0.1 ^b^	3.96 ± 0.1 ^a^	3.55 ± 0.38 ^a^
	42	Acetone	/	/	6.97 ± 2.54 ^a^
	43	5-Methyl-4-hexen-3-one	/	2.96 ± 0.32 ^a^	/
	44	3-Ethylcyclopentanone	/	/	2.94 ± 0.15 ^a^
	45	Acetophenone	2.56 ± 0.34 ^a^	/	2.87 ± 0.26 ^a^
	46	3-Octen-2-one	2.53 ± 0.15 ^a^	/	1.55 ± 0.18 ^b^
	47	4-Nonanone	/	1.54 ± 0.04 ^a^	1.55 ± 0.24 ^a^
	48	2,5-Dimethyl-3-hexanone	0.36 ± 0.2 ^a^	0.34 ± 0.1 ^a^	/
	49	2,2,5-Trimethylhexane-3,4-dione	0.88 ± 0.16 ^a^	/	/
Heterocycles	50	2-Pentylfuran	817.05 ± 95.09 ^b^	833.04 ± 37.86 ^b^	1024.09 ± 37.76 ^a^
	51	2-(1-Pentenyl)furan	2.89 ± 0.58 ^a^	3.33 ± 0.21 ^a^	3.82 ± 0.95 ^a^
	52	2-Hexylfuran	1.85 ± 0.22 ^a^	/	/
Terpenes	53	3-Methyl-1-hexene	/	0.83 ± 0.01 ^a^	
	54	9-Methyl-1-undecene	1.03 ± 0.12 ^a^	/	/
Others	55	1H-Indene, octahydro-2,2,4,4,7,7-hexamethyl-, trans-	9.48 ± 1.25 ^a^	/	9.33 ± 3.35 ^a^
	56	Naphthalene	3.46 ± 0.54 ^a^	3.01 ± 0.14 ^a^	3.3 ± 1 ^a^
	57	1,2,4,5-Tetramethylbenzene	2.32 ± 0.19 ^b^	/	3.17 ± 0.24 ^a^
	58	1H-Tetrazol-5-amine	/	0.24 ± 0.08 ^a^	0.27 ± 0.03 ^a^
	59	Semicarbazide	/	/	2.89 ± 2.25 ^a^
	60	(1-Ethylpropyl)benzene	/	/	2.69 ± 0.35 ^a^
	61	N-Isobutyl(phenyl)methanesulfonamide	/	/	2.74 ± 0.56 ^a^
	62	2,4,5-Trimethyl-1,3-dioxolane	/	1.72 ± 0 ^a^	/
	63	1,1,4a,5,6-Pentamethyldecahydronaphthalene	/	0.96 ± 0 ^a^	/
	64	2-Acetylthiazole	0.81 ± 0.05 ^a^	/	/
	65	2-Methoxy phenol	0.59 ± 0.13 ^a^	/	/
	66	Dihexyverine	/	0.23 ± 0.05 ^a^	/
	67	Ethoxyethene	/	0.2 ± 0.04 ^a^	/

Note: different superscript letters means significantly different (*p* < 0.05).

**Table 2 foods-14-01471-t002:** Partial differential metabolites between co-culture and pure culture of *A. pasteurianus* LA10.

Metabolite	Fold Change	Regulate	VIP Value
8-Hydroxyguanine	0.7166	down	8.2403
3-Hydroxy-4-methoxyphenyllactic acid	1.2953	up	6.8915
Phenyllactic acid	1.1295	up	6.3087
Xanthine amine	0.8554	down	6.2165
Franguloside	1.1626	up	6.0155
S-Adenosylhomocysteine	1.1013	up	4.7833
Phenylpyruvic acid	0.9111	down	4.4769
5′-Methylthioadenosine	0.9299	down	4.1068
(E)-10-Hydroxy-8-decenoic acid	1.0675	up	4.0472
Ketoleucine	0.9479	down	3.9116
Enniatin B	1.0441	up	3.8112
4-Vinylphenol	1.0659	up	3.7345
Ethyl 3-hydroxydodecanoate	1.0993	up	3.7238
1,4,7,10,13,16-Hexaoxacyclooctadecane	1.0604	up	3.4713
6-Hydroxyhexanoic acid	1.0352	up	3.1451
Uric acid	0.9635	down	3.134
5-Chloro-2′-deoxyuridine	1.0459	up	3.1175
(2S,3R,4S,5R,6R)-6-Ethyloxane-2,3,4,5-tetrol	1.0401	up	3.0507
PC(18:2(9Z,12Z)/18:2(9Z,12Z))	0.9675	down	3.016
5-Methyl-2-furancarboxaldehyde	0.9615	down	2.9399
Enniatin B1	1.0264	up	2.9028
Epitiostanol	1.0326	up	2.8945
1-Hexanol	1.0388	up	2.7946

**Table 3 foods-14-01471-t003:** Partial differential metabolites between co-culture and pure culture of *L. amylovorus* LL34.

Metabolite	Fold Change	Regulate	VIP Value
L-Asparagine	0.5834	down	8.7843
Scopoletin	1.7944	up	8.6923
Iminodiacetic acid	0.6175	down	8.0915
3a,7a-Dihydroxy-5b-cholestane	1.4394	up	7.2836
4-Oxo-2-azetidinecarboxylic acid	0.7232	down	7.162
L-Aspartic acid	0.7397	down	6.8104
Ureidopropionic acid	0.6903	down	6.8023
Aspartic acid	0.7887	down	6.5172
Isofraxidin	1.3725	up	6.3634
3-Indolebutyric acid	1.2467	up	6.0585
2-(Carbamoylamino)propanoic acid	0.8051	down	5.9903
5-Hydroxymethyl-2-furancarboxaldehyde	1.2321	up	5.7925
MALEAMIC ACID	0.7766	down	5.659
Tolmetin	1.1964	up	5.379
4-Hydroxyvalproic acid	1.1994	up	5.2886
Cyclopropyl–methoxycarbonyl metomidate	1.1838	up	5.281
DG(22:4(7Z,10Z,13Z,16Z)/15:0/0:0)	1.1599	up	5.1935
Franguloside	1.1716	up	5.142
3-Ethyl-5-hydroxy-4,5-dimethyl-pyrrolin-2-one	1.1865	up	5.0692
1,2-Dilinolenoyl-3-(4-aminobutyryl)propane-1,2,3-triol	1.1516	up	4.8926
3-Methoxyphenol sulfate	1.175	up	4.7342
2-Succinylbenzoate	1.1424	up	4.3289
N-lactoyl-Methionine	1.127	up	4.3233
5-Hydroxyvalproic acid	1.1097	up	4.245
S-Adenosylhomocysteine	1.1109	up	4.1535

## Data Availability

The data are provided in the text.

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
