# Peer review of "Interaction Between Lactic Acid Bacteria and Acetic Acid Bacteria in Sichuan Bran Vinegar: Impact on Their Growth and Metabolites"

_foods, 2025, doi:10.3390/foods14091471_

Round 1
Reviewer 1 Report
Comments and Suggestions for Authors
According to the title, the work should concern interaction between LAB and AAB in Sichuan bran vinegar with particular emphasis on growth and metabolites. In my opinion, the title does not reflect the actual content of the work. I haven't comments on the scope of the study and interpretation of results regarding the determination of the vineger composition diversity. However, I have comments on the microbiological part. The results obtained from the microbiological part are limited only to description. In my opinion, the publication lacks an interpretation of the results and a scientific explanation of the mechanisms of these changes and the effects resulting from them. The short chapter 5.3. is only an attempt at it. Regarding the title, I expected the authors to analyze the mechanisms of metabolic regulation on a scientific level. This level of analysis of results and discussion is unsatisfactory. There are scientific publications available on this topic.
In addition, I have a few specific comments:
- Introduction "As is well known, the microbial composition of vinegar is complex, with microbial interactions being essential for its stability and functionality". In my opinion, this sentence is directly related to the topic of the paper and should be expanded. The authors do not mention what other microorganisms are isolated from vinegar and what their significance is for the fermentation process and the obtained product.
- Strains and growth conditions (2.1.). As in the case of the composition of the culture medium for AAB, the composition for LAB should be given as follows
-
Solid-state fermentation and sampling (2.2.1). The authors inoculated the medium with L. amylovorus LL34 (106 CFU/g) and A. pasteurianus LA10 (4×105 CFU/g). Why is there a difference of up to one log? Is this methodologically justified and could it have influenced the results obtained?
The bacteria L. amylovorus and A. pasteurianus belong to different metabolic groups. Under certain conditions they can form consortia and synchronize the processes of substrate transformation, an example of which is SCOBY. What is the methodological assumption that MRS medium is a suitable culture medium for AAB. The GYE medium was used in point 2.1. for AAB - Enumeration of strains. What is the methodological assumption that MRS medium is a suitable culture medium for AAB (as in 2.1.)
According to the literature, A. pasteurianus is an aerobic bacterium. In the case of L. amylovorus, the cultivation process is usually carried out in a limited amount of oxygen. Please justify (e.g. with reference to a publication) that MRS agar was used to determine the number of AAB and LAB bacteria, and the cultivations were also carried out in aerobic conditions.
Author Response
Response to Reviewer 1 Comments
Comments1:[Introduction "As is well known, the microbial composition of vinegar is complex, with microbial interactions being essential for its stability and functionality". In my opinion, this sentence is directly related to the topic of the paper and should be expanded. The authors do not mention what other microorganisms are isolated from vinegar and what their significance is for the fermentation process and the obtained product.]
Response 1:Thank you for the suggestion, we have modified Introduction [The dominant bacteria in the solid-state fermentation of vinegar differ at different periods, such as mold in the early stage of fermentation (starch saccharification fermentation), yeast in the middle stage (alcohol fermentation), and acetic acid bacteria in the late stage (acetic acid fermentation). At the same time, there are other dominant bacteria, such as Lactobacillus, Bacillus, Stenotrophomonas, Methyloversatilis, and Amycolatopsis, among which lactic acid can be produced to neutralize the irritation of acetic acid [11]. Bacillus can secrete a variety of hydrolases, which play an important role in liquefaction and saccharification, and can produce organic acids through tricarboxylic acid cycle [12]]
Comments 2:[1.Strains and growth conditions (2.1.). As in the case of the composition of the culture medium for AAB, the composition for LAB should be given as follows]
Response 2:Thank you for the suggestion, we have modified it [MRS broth (10g Peptone, 8g Beef Extract, 4g Yeast Extract, 20g glucose, 2g K₂HPO₄, 5g CH₃COONa, 2g Ammonium Citrate, 0.2g MgSO₄·7H₂O, 0.05g MnSO₄·H₂O, 1ml Tween 80, pH 6.2-6.5, 1L H2O]
Comments 3:[Solid-state fermentation and sampling (2.2.1). The authors inoculated the medium with L. amylovorus LL34 (106 CFU/g) and A. pasteurianus LA10 (4×105 CFU/g). Why is there a difference of up to one log? Is this methodologically justified and could it have influenced the results obtained? The bacteria L. amylovorus and A. pasteurianus belong to different metabolic groups. Under certain conditions they can form consortia and synchronize the processes of substrate transformation, an example of which is SCOBY. What is the methodological assumption that MRS medium is a suitable culture medium for AAB. The GYE medium was used in point 2.1. for AAB]
Response 3: Thank you for the suggestion. Previous research found that the concentration difference between L. amylovirus and A. Pasteurianus in the vinegar solid-state fermentation process was 2.5 times, so we chose this concentration to inoculate fortified Cupei to simulate the real fermentation system. During the pre-experiment, we attempted to inoculate both strains into the Cupei medium of MRS or GYE broth. The results showed that the two strains grew well in the Cupei stimulation medium (Wheatbran 30%, Wheatflower 20%, and Mrs broth 50%); therefore, we chose this medium.
Comments 4:[Enumeration of strains. What is the methodological assumption that MRS medium is a suitable culture medium for AAB (as in 2.1.)
According to the literature, A. pasteurianus is an aerobic bacterium. In the case of L. amylovorus, the cultivation process is usually carried out in a limited amount of oxygen. Please justify (e.g. with reference to a publication) that MRS agar was used to determine the number of AAB and LAB bacteria, and the cultivations were also carried out in aerobic conditions.]
Response 4: We thank the reviewer for their constructive suggestions. For bacterial cultivation, we employed a Cupei stimulation medium composed of wheat bran (30%), wheat flour (20%), and MRS broth (50%). Preliminary experiments confirmed that both L. amylovorus LL34 and A. pasteurianus LA10 exhibited robust growth in this medium. To maintain adequate oxygenation in the solid-state culture system, we implemented regular substrate turnover during the cultivation process.
Seed cultures of the two strains were prepared as follows. L.amylovorus LL34 was cultured in MRS broth to achieve an inoculum density of 106 CFU/g. A. pasteurianus LA10 was propagated in GYE broth, yielding an inoculum of 4×105 CFU/g.
As demonstrated in Section 3.1, both bacterial strains exhibited robust growth profiles in pure or mixed culture conditions using MRS medium. These experimental findings are further supported by relevant literature citations incorporated into the text. We appreciate the reviewer’s insightful comments, which have strengthened our methods.
Reviewer 2 Report
Comments and Suggestions for Authors
The introduction (lines 1–69) establishes vinegar fermentation's cultural and biochemical importance and emphasizes the novelty of dissecting interspecies interactions at the strain level. The methodology (lines 70–170) is rigorous and includes co-culture fermentation, enumeration, metabolite quantification via HPLC, and advanced analytical techniques (GC–MS and untargeted UHPLC-MS/MS). The results (lines 171–478) are well-structured, showing that while A. Pasteurianus LA10 maintained robust growth; its acidogenic and metabolic output (e.g., acetic acid, volatiles) was suppressed in co-culture, and LL34’s growth was inhibited (lines 175–210). The multi-omic profiling revealed shifts in both volatile (lines 211–312) and non-volatile (lines 313–478) metabolomes, enriching amino acid and phenylpropanoid pathways. Key differential metabolites such as phenyllactic acid and 4-vinylphenol, known for antimicrobial and flavor properties, underscore the functional implications of microbial competition. However, the discussion could benefit from deeper mechanistic insights and broader contextualization within existing microbial ecology literature. Additionally, transcriptomic or proteomic validation is suggested in the conclusions (lines 479–542) but could be framed more assertively as a future direction.
Author Response
Response to Reviewer 2 Comments
Comments1:[The introduction (lines 1–69) establishes vinegar fermentation's cultural and biochemical importance and emphasizes the novelty of dissecting interspecies interactions at the strain level. The methodology (lines 70–170) is rigorous and includes co-culture fermentation, enumeration, metabolite quantification via HPLC, and advanced analytical techniques (GC–MS and untargeted UHPLC-MS/MS). The results (lines 171–478) are well-structured, showing that while A. Pasteurianus LA10 maintained robust growth; its acidogenic and metabolic output (e.g., acetic acid, volatiles) was suppressed in co-culture, and LL34’s growth was inhibited (lines 175–210). The multi-omic profiling revealed shifts in both volatile (lines 211–312) and non-volatile (lines 313–478) metabolomes, enriching amino acid and phenylpropanoid pathways. Key differential metabolites such as phenyllactic acid and 4-vinylphenol, known for antimicrobial and flavor properties, underscore the functional implications of microbial competition. However, the discussion could benefit from deeper mechanistic insights and broader contextualization within existing microbial ecology literature. Additionally, transcriptomic or proteomic validation is suggested in the conclusions (lines 479–542) but could be framed more assertively as a future direction.]
Response 1: [We appreciate the reviewer's valuable suggestion. In our subsequent research, we employed integrated metagenomic and transcriptomic analyses to elucidate the synergistic mechanisms between these two bacterial strains. These comprehensive findings will be presented in detail in a separate publication currently in preparation.]
Round 2
Reviewer 2 Report
Comments and Suggestions for Authors
The manuscript has been thoroughly revised to incorporate all recommended changes, enhancing clarity, methodological rigor, and scientific impact. The introduction clearly defines the research gap, while the methodology provides justifications and ensures reproducibility. The discussion has been expanded for deeper analysis, addressing key findings, limitations, and comparisons with the literature. The conclusion now emphasizes broader implications, scalability, and future research. Terminology has been standardized, redundancies removed, and recent references integrated. With these comprehensive improvements, the manuscript meets publication standards, presenting a well-structured and scientifically robust contribution suitable for dissemination in its current form.